# Metagenome and Resistome Analysis of Beta-Lactam-Resistant Bacteria Isolated from River Waters in Surabaya, Indonesia

**DOI:** 10.3390/microorganisms12010199

**Published:** 2024-01-18

**Authors:** Ryohei Nomoto, Kayo Osawa, Shohiro Kinoshita, Koichi Kitagawa, Noriko Nakanishi, Rosantia Sarassari, Dadik Raharjo, Masato Fujisawa, Kuntaman Kuntaman, Toshiro Shirakawa

**Affiliations:** 1Department of Infectious Diseases, Kobe Institute of Health, Kobe 650-0046, Japan; ryohei_nomoto@office.city.kobe.lg.jp (R.N.); noriko_nakanishi@office.city.kobe.lg.jp (N.N.); 2Department of Medical Technology, Kobe Tokiwa University, Kobe 653-0838, Japan; 3Division of Advanced Medical Science, Kobe University Graduate School of Science, Technology and Innovation, Kobe 650-0017, Japan; kino@med.kobe-u.ac.jp (S.K.); kkitagawa@port.kobe-u.ac.jp (K.K.); toshiro@kobe-u.ac.jp (T.S.); 4Department of Microbiology, Faculty of Medicine, Airlangga University, Surabaya 60132, Indonesia; santisarassari@yahoo.com (R.S.); kuntaman@fk.unair.ac.id (K.K.); 5Institute of Tropical Disease, Airlangga University, Surabaya 60286, Indonesia; dadik.tdc@gmail.com; 6Division of Urology, Department of Organ Therapeutics, Faculty of Medicine, Kobe University Graduate School of Medicine, Kobe 650-0017, Japan; masato@kobe-u.ac.jp

**Keywords:** antimicrobial-resistant bacteria (AMR), environmental river, whole genome sequencing (WGS), antimicrobial resistance genes (ARGs), public health

## Abstract

Antimicrobial agents are administered to humans and livestock, and bacterial antimicrobial resistance (AMR) and antimicrobial agents are released into the environment. In this study, to investigate the trend of AMR in humans, livestock, and the environment, we performed a metagenomic analysis of multidrug-resistant bacteria with CHROMagar ESBL in environmental river water samples, which were collected using syringe filter units from waters near hospitals, downtown areas, residential areas, and water treatment plants in Surabaya, Indonesia. Our results showed that *Acinetobacter*, *Pseudomonas*, *Aeromonas*, *Enterobacter*, *Escherichia*, and *Klebsiella* grew in CHROMagar ESBL; they were most frequently detected in water samples from rivers surrounding hospitals contaminated with various AMR genes (ARGs) in high levels. These results identified bacteria as ARG reservoirs and revealed that hospitals could be sources for various ARGs disseminated into the environment. In conclusion, this study details a novel metagenomic analysis of collected bacteria in environmental water samples using a syringe filter unit for an AMR epidemiological study based on the One Health approach.

## 1. Introduction

According to an antimicrobial resistance (AMR) review by the UK government in 2014 [1], the number of annual deaths owing to bacterial AMR will increase to 10 million by 2050 if appropriate measures are not taken. Another predictive statistical method estimated 4.95 and 1.27 million deaths associated with and attributable to bacterial AMR worldwide in 2019, respectively [2]. As antimicrobial agents are administered to humans as well as livestock, and bacterial AMR and antimicrobial agents are released into the environment, adopting a One Health approach is increasingly important to investigate trends in AMR that cover humans, livestock, and the environment [3,4,5]. In Indonesia, the One Health approach is essential for investigating AMR trends because inexpensive antimicrobial agents are over used, sewage infrastructure is inadequate, and poultry consumption is high.

Recent advances in genetic analysis technologies, such as next-generation sequencing, have enabled whole-genome sequencing of bacteria, enabling the investigation of the molecular epidemiology of AMR genes (ARGs) and bacterial AMR in further detail [6,7]. In addition, metagenomic analysis using next-generation sequencing allows a comprehensive investigation of the microbiota in humans, livestock, and the environment (e.g., river water) [8]. Because the administration of antimicrobial agents and their leakage into the environment directly affects the ARG profile (resistome) in humans and livestock, comprehensively analyzing ARGs in the environment will help to clarify the mechanism of AMR spread [7].

In this study, we performed a metagenome and resistome analysis of multidrug-resistant bacteria stored in syringe filter units from environmental river water samples located near hospitals, downtown areas, residential areas, and water treatment plants in Surabaya, Indonesia. Employing metagenome and resistome analyses for analyzing the environmental contaminating bacteria, especially in river water, enabled the creation of a novel comprehensive molecular epidemiological method under the One Health approach. This method is expected to identify ARG reservoirs and transmission vehicles in the environment and provide new insights into the molecular mechanisms of AMR spread. 

## 2. Materials and Methods

### 2.1. Sample Collection and Culture Condition

River water samples were collected from six locations in the urban areas of Surabaya, Indonesia, in December 2019 (nearby; (1) Treatment Plant: TP, (2) Galaxy-Mall (downtown): GM, (3) Klampus (residential area): KL, (4) Kalibokor (residential area): KA, (5) Haji-Hospital: HH, (6) Soetomo-Hospital: SH; Figure 1). To trap and collect bacteria, 30 mL water samples were filtered with a Millex-GV Syringe Filter Unit, 0.22 µm pore size, PVDF-33 mm membrane, and stored using the Syringe Filter Unit (Milliporesigma, Bedford, MA, USA). The syringe filters were attached to an empty 10 mL syringe, and 5 mL phosphate-buffered saline (PBS) was aspirated to suspend the bacteria adsorbed on the filter. The aspirate was collected in a 15 mL tube and centrifuged at 10,000× *g* for 10 min at 4 °C. The supernatant was discarded and the pellet was resuspended in PBS (0.5 mL). Then, 200 µL of the suspension was seeded into CHROMagar ESBL plates (CHROMagar, Paris, France) and cultured overnight at 37 °C. After incubation, 2 mL of PBS was added to the CHROMagar ESBL plates and all colonies were collected. Of the collected bacterial suspension, 200 µL was centrifuged and the pellet was resuspended in 1 mL PBS. DNA extraction was performed on 200 µL of the resuspension solution and metagenomic DNA was extracted using the Nucleospin Tissue Kit (MACHEREY-NAGEL, Düren, Germany), as per manufacturer’s instructions.

### 2.2. Metagenomic DNA-Seq Analysis of Extended Spectrum Beta-Lactamase (ESBL)-Producing Bacteria

A metagenomic DNA-seq library was prepared using the QIAseq FX DNA library kit (QIAGEN, Hilden, Germany), and sequencing was performed using the Illumina MiSeq system (Illumina, San Diego, CA, USA) with v3 chemistry (2 × 300-bp format). Adapter and low-quality sequences in short-read data were trimmed using fastp v0.23.2 [9] with default settings. Metagenomic analysis was performed using the MePIC software v2.0 [10] with default parameters, and the results were visualized using MEGAN [11] to determine the distribution of operational taxonomic units (OTUs). 

### 2.3. Resistome Analysis

ARGs were predicted using the Resistance Gene Identifier (RGI v5.2.0; default parameters of RGI bwt mode) and the Comprehensive Antibiotic Resistance Database (card v3.1.4) [12]. OTUs of ARGs in the database were created via clustering at ≥90% sequence identity and ≥80% coverage. The detected genes were summarized for each OTU. For normalization, per kilobase per million mapped reads (RPKM) were calculated using the following formula: 

RPKM = number of detected reads against OTUs/(average gene length of the detected OTUs (bp) × total number of trimmed reads) × 10^9^. 

## 3. Results

### 3.1. Bacterial Proportion of Strains Growing on CHROMagar ESBL Plate from Indonesian River Water

The taxonomic classification of cultured bacteria from Indonesian river water samples on CHROMagar ESBL plates at the genus level is shown in Figure 2. *Acinetobacter* and *Pseudomonas* were the most frequent bacterial genera that grew in CHROMagar ESBL from the water collected at all sampling sites. Bacterial genera differed among the samples. The bacterial proportion of the river water sample from TP was the poorest in diversity, with *Acinetobacter* (71.5%) and *Pseudomonas* (27.9%) being the dominant genera. *Acinetobacter* and *Pseudomonas* were detected in river water samples from GM, KL, and KA, with high abundance ratios of 75.6%, 68.0%, and 57.3%, respectively, whereas in samples from HH and SH surrounding the hospital, the abundance ratios of both species were comparatively lower at 28.9% and 31.8%, respectively. Bacterial genera, such as *Acinetobacter*, *Pseudomonas*, *Aeromonas*, *Escherichia*, *Enterobacter*, and *Klebsiella* were the dominant constituents in river water culture samples from HH and SH, and the representative ESBL-producing bacteria, *Escherichia*, *Enterobacter*, and *Klebsiella*, had a higher proportion ratio than at the other sampling sites (Figure 2 and Appendix A).

### 3.2. ARG Proportion of Growing on CHROMagar ESBL Plate from Indonesian River Water

The metagenomic DNA-seq short reads were analyzed using RGI software v5.1.0 in the CARD AMR gene database under normalization using reads per kilobase of exon per million mapped reads (RPKM). ARGs to quaternary aminoglycoside (AMG), β-lactam, quaternary ammonium compounds (QAC), sulfonamide (SA), and macrolide were detected substantially in culture samples from all water collection points (Figure 3). Resistome analysis showed that drug resistance genes for aminoglycoside and β-lactam antibiotics were predominantly detected in all samples. The most abundant ARGs in the total RPKMs were detected in the river water samples from SH, followed by samples from HH. In contrast, river water from TP had the lowest total RPKM values, and the ARGs detected were mostly derived from *Acinetobacter* and *Pseudomonas*, with limited diversity (Figure 3). 

ARGs detected in each AMR category were further classified as orthologous genes (or gene families) (Figure 4 and Appendix A). The SA resistance gene *sul1* was most frequently detected at 660.9 RPKM (Appendix A); *sul2* detection was slightly lower at 398.7 RPKM, whereas that of *sul3* was much lower at 29.1 RPKM. The second most detected ARG was the QAC resistance gene *qacEΔ1* at 643.8 RPKM (Appendix A), suggesting that basic gene sets (*sul1* and *qacEΔ1*) in the class 1 integron [13] were predominant in the detected ARGs. Multiple AMG resistance genes were identified in the clinically important ARGs (Figure 4). The four most predominant AMG resistance genes belonged to the *aac* and *aph* gene families (Appendix A), which contribute to streptomycin resistance, implying that streptomycin could be the most potent selective agent among aminoglycosides. Following this, high RPKM of *kdpE* and *acr*, efflux pumps located on the chromosomes of *E. coli*, were detected. Regarding the β-lactamase gene type, *bla*_OXA_, and *bla*_TEM_ were the most considerably detected (Figure 4), followed by POM-1 and ADC, which were present in *Pseudomonas* and *Acinetobacter*, respectively. The *bla*_CTX-M_ cluster, an ESBL gene commonly detected in clinical settings, was most predominant in river water culture samples around hospitals such as HH and SH. In particular, the *bla*_NDM_ gene cluster was also detected in samples from SH (Figure 4, Appendix A).

In addition, *mcr* genes contributing to colistin resistance were detected in KA and HH river water samples with extremely low RPKM values.

## 4. Discussion

In this study, a metagenomic analysis was conducted on bacterial strains collected from river water using syringe filter units and cultured in CHROMagar ESBL. In developing countries such as Indonesia, the widespread use of antimicrobials has become a public health problem because their components and antimicrobial-resistant bacteria are released into environmental rivers; however, only few studies report on these issues [14]. Filtering large quantities of water using specialized filtration equipment is necessary to investigate the presence of antimicrobial-resistant bacteria in rivers. In particular, investigating antimicrobial-resistant bacteria in areas lacking well-developed transportation is challenging. Therefore, we determined that environmental water could be easily collected using syringe filter units and that bacteria could be easily recovered from the units. Previously, the sensitivity of CHROMagar ESBL medium for the detection of ESBL-producing *Enterobacteriaceae* was evaluated using spiked stools, and 93.9% of the sensitivity of the CHROMagar ESBL medium was confirmed [15]. In addition, Vurayai and his colleagues investigated the environmental ESBL bioburden in neonatal intensive care units (NICU) using CHROMagar ESBL, and they concluded that the CHROMagar ESBL is useful to identify ESBL reservoirs and transmission vehicles [16]. Similarly, in our study, metagenomic analysis of bacteria collected via syringe filter units and screened using the CHROMagar ESBL medium revealed reservoirs and transmission vehicles of ESBL and various ARGs in rivers near urban and residential areas, water treatment plants, and hospitals. 

Throughout this analysis, *Acinetobacter* and *Pseudomonas* were the most frequently detected bacterial genera; they were even isolated from water samples of treatment plants, which are treated water sources (Figure 2). Multidrug-resistant *Acinetobacter baumannii* and *Pseudomonas aeruginosa* often cause infections that are difficult to treat. The isolation of these organisms from drinking water sources should be alarming [17,18]. *Enterobacteriaceae,* including *Escherichia, Enterobacter*, and *Klebsiella*, were also detected at high frequencies in rivers and wastewater, particularly near hospitals (Figure 2). ESBL-producing *E. coli* is a multidrug-resistant organism targeted by the WHO’s Tricycle Project based on the One Health approach and should continue to be closely monitored [19]. In this study, *Aeromonas* growing on CHROMagar ESBL was also detected in most water samples (Figure 2). Recently, *Aeromonas* carrying the *bla-_NDM-1_* gene has been reported to cause community-acquired pneumonia, suggesting that *Aeromonas* plays an important role in the spread of multidrug-resistant bacteria as a reservoir of ARGs in the community and environment [20]. 

In this study, resistome analysis revealed that drug-resistant genes to antibiotics commonly used in clinical practice were detected at a high frequency with a high diversity of bacterial proportions and high total RPKM values in river water around hospitals (Figure 2 and Figure 3). In addition, ARGs that pose a serious clinical threat, such as *NDM* and *MCR*, were detected in rivers near hospitals (Figure 4 and Appendix A). ARG effluents from these facilities have a considerable impact on the surrounding environment. The use of antimicrobial agents is the primary cause of AMR epidemics and worldwide consumption of antimicrobial agents reportedly increased by 65% between 2000 and 2015 [21]. Currently, proper usage of antimicrobial agents is strictly required in healthcare settings, and antimicrobial use in developed countries is decreasing; however, in emerging countries, their use continues to increase rapidly [22]. Antimicrobial abuse is the major causative factor of the spread of AMR, as exposure to antimicrobial agents increases selection pressure for AMR and promotes the horizontal spread of ARGs [23].

## 5. Conclusions

In conclusion, a novel metagenomic and resistome analysis of environmental waters using CHROMagar ESBL revealed that *Acinetobacter*, *Pseudomonas*, *Aeromonas*, *Enterobacter*, *Escherichia*, and *Klebsiella* were important reservoirs and spreaders of ARGs, and water surrounding hospitals was contaminated with various ARGs at high levels. This novel metagenomic analysis of environmental water could be a powerful tool for AMR molecular epidemiological studies based on the One Health approach.

## Figures and Tables

**Figure 1 microorganisms-12-00199-f001:**
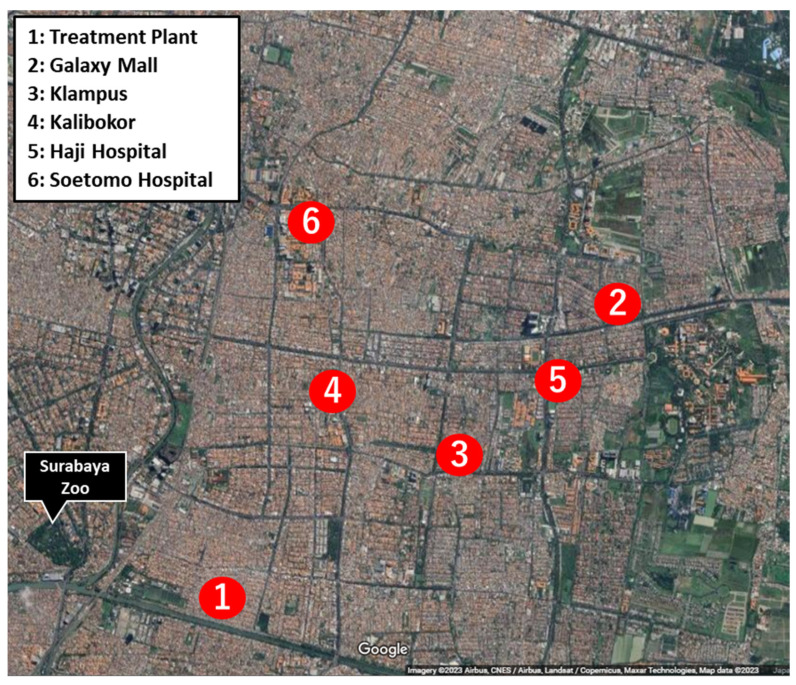
Map of water sampling sites in the Surabaya area. Surabaya Zoo was marked as a landmark.

**Figure 2 microorganisms-12-00199-f002:**
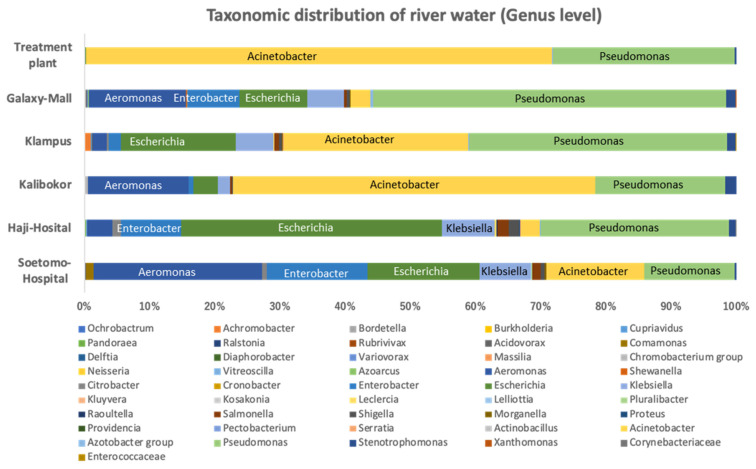
Taxonomic classification at the genus level of river water culture samples on CHROMagar ESBL plates based on metagenomic DNA-Seq analysis results.

**Figure 3 microorganisms-12-00199-f003:**
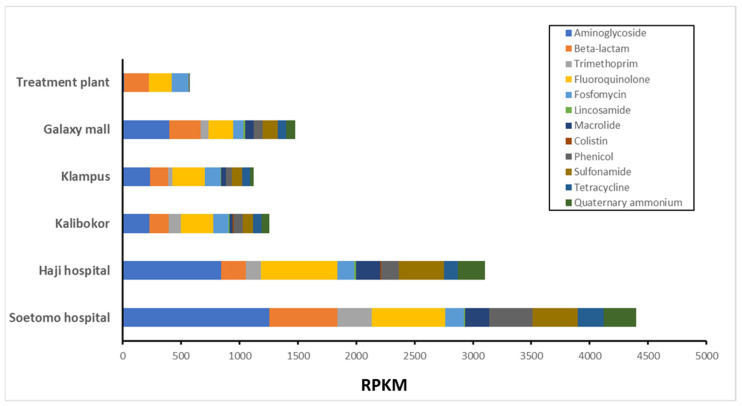
Metagenomic analysis for AMR genes detection for each culture sample of river water. Metagenomic DNA-seq short reads were analyzed using RGI software v5.1.0 in CARD database, followed by normalization with RPKM. AMR, antimicrobial resistance; RPKM, per kilobase per million mapped reads.

**Figure 4 microorganisms-12-00199-f004:**
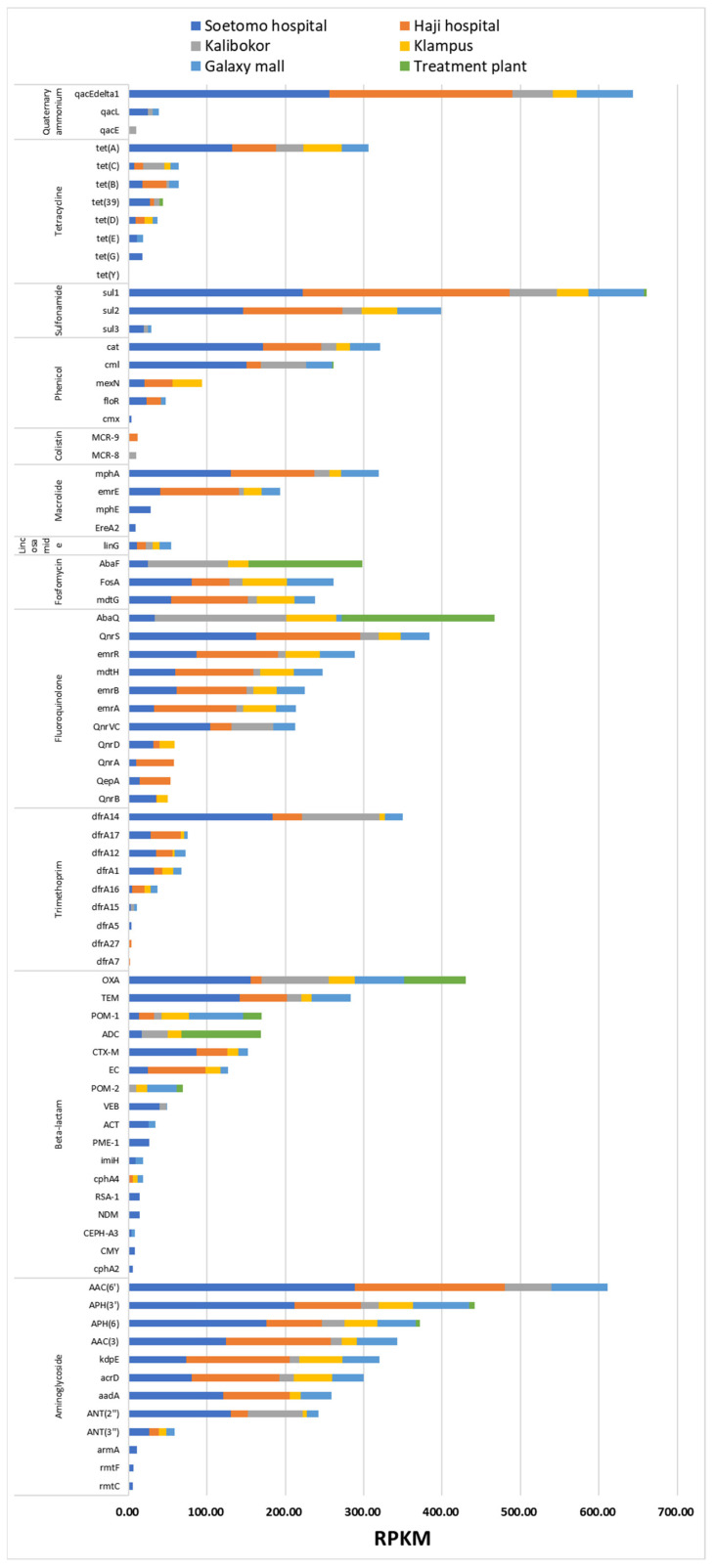
AMR gene profiling based on AMR categories. Trends in AMR gene variations; detected genes in each AMR category at every sampling site. AMR, antimicrobial resistance.

## Data Availability

All metagenome read data obtained as part of this study were deposited in DDBJ/EMBL/GenBank under the BioProject accession number PRJDB16871. The DRA accession numbers were DRR513858 (TP), DRR513859 (GM), DRR513860 (KL), DRR513861 (KA), DRR513862 (HH), and DRR513863.

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
