# Peer review of "Metagenome and Resistome Analysis of Beta-Lactam-Resistant Bacteria Isolated from River Waters in Surabaya, Indonesia"

_microorganisms, 2024, doi:10.3390/microorganisms12010199_

Round 1
Reviewer 1 Report
Comments and Suggestions for Authors
In this study, a metagenomic analysis was conducted on bacterial strains collected from river water using syringe filter units and cultured in CHROMagar ESBL. Here, the metagenome analysis of bacteria screened by CHROMOagar ESBL medium revealed reservoirs and transmission vehicles of ESBL and various ARGs in rivers nearby urban and residential areas, water treatment plants, and hospitals.
Overall, this novel metagenomic analysis of environmental water could be a powerful tool for AMR molecular epidemiological studies based on the One Health Approach. I think it can be accepted once some of my questions are fully addressed.
1. Many duplicate contents. For instance. Page 7, line 186-192. There are two identical sentences! Please double check the writing the proofread it before resubmitting the revised article.
2. Too many grammar issues in the main text. Page 2-“...comprehensively analyzing ARGs in the environment will clarifying the mechanism of AMR spread.” It should be “clarify” here.
3. I double checked Figure 2, it looks quite confusing. I recommend to add a supplementary table for readers, which is better to show the detailed value for each group.
Comments on the Quality of English Language
Need to be improved, please check my comments.
Author Response
Comments and Suggestions for Authors
In this study, a metagenomic analysis was conducted on bacterial strains collected from river water using syringe filter units and cultured in CHROMagar ESBL. Here, the metagenome analysis of bacteria screened by CHROMOagar ESBL medium revealed reservoirs and transmission vehicles of ESBL and various ARGs in rivers nearby urban and residential areas, water treatment plants, and hospitals.
Overall, this novel metagenomic analysis of environmental water could be a powerful tool for AMR molecular epidemiological studies based on the One Health Approach. I think it can be accepted once some of my questions are fully addressed.
- Many duplicate contents. For instance. Page 7, line 186-192. There are two identical sentences! Please double check the writing the proofread it before resubmitting the revised article.
Response: Thank you for the comment. We delete it.
In addition, Vurayai and his colleagues investigated the environmental ESBL bioburden in the neonatal intensive care units (NICU) by using CHROMagar ESBL, and they concluded that the CHROMagar ESBL is useful to identify ESBL reservoirs and transmission vehicles [16]. Similarly, in our study, metagenomic analysis of bacteria collected via syringe filter units and screened using the CHROMagar ESBL medium revealed reservoirs and transmission vehicles of ESBL and various ARGs in rivers near urban and residential areas, water treatment plants, and hospitals.
(Page 7, line 183 to 187).
- Too many grammar issues in the main text. Page 2-“...comprehensively analyzing ARGs in the environment will clarifying the mechanism of AMR spread.” It should be “clarify” here.
Response: Thank you for the comment. We correct it.
…comprehensively analyzing ARGs in the environment will clarify the mechanism of AMR spread [7].
(Page 2, line 54).
- I double checked Figure 2, it looks quite confusing. I recommend to add a supplementary table for readers, which is better to show the detailed value for each group.
Response: Thank you for the comment. We double checked Figure 2, it looks quite confusing. I recommend to add a supplementary table for readers (Table S1), which is better to show the detailed value for each group.
…the representative ESBL-producing bacteria, Escherichia, Enterobacter, and Klebsiella, had a higher proportion ratio than at the other sampling sites (Fig. 2 and Table S1).
(Page 4, line 123).
Table S1: Summary of OTUs detected in CHROMagar ESBL culture samples from Indonesian river water.
(Page 8, line 224 to 225).

Reviewer 2 Report
Comments and Suggestions for Authors
The extension of public health surveillance of antibiotic resistance from the hospital to the river by metagenomics represents an interesting perspective, not yet largely used (I was not aware of it until reading your paper and the cited references), by contrast to SARS-CoV-2 river investigation becoming a standard of epidemiological surveillance.
The paper is clearly presented, and very detailed. Difficult to shorten without losing information.
I have nevertheless some discomfort with the mass of data: on your Figure 1, for example, there are so many microorganisms identified by metagenomics that it is impossible for a reader to see what is of clinical relevance in human or in veterinary medicine.
Difficult to modify this very extensive data analysis without losing major information.
So, it is preferable to publish your paper in its present form, but an effort should be done in the next future of your research to try to put more in evidence a direct relationship between clinical cases of antibiotic resistance in humans and animals.
This is a general problem of artificial intelligence: the machine produces outputs which cannot be mastered in a pertinent way by humans. So, now that you have a potential interesting and promising technology, the future is open for researchers restricting their vision to very precise and most pertinent questions in infectious diseases.
This is the reason why I indicate "not applicable" or "no answer" in the appreciation of your manuscript.
Author Response
The extension of public health surveillance of antibiotic resistance from the hospital to the river by metagenomics represents an interesting perspective, not yet largely used (I was not aware of it until reading your paper and the cited references), by contrast to SARS-CoV-2 river investigation becoming a standard of epidemiological surveillance.
The paper is clearly presented, and very detailed. Difficult to shorten without losing information.
I have nevertheless some discomfort with the mass of data: on your Figure 1, for example, there are so many microorganisms identified by metagenomics that it is impossible for a reader to see what is of clinical relevance in human or in veterinary medicine.
Difficult to modify this very extensive data analysis without losing major information.
So, it is preferable to publish your paper in its present form, but an effort should be done in the next future of your research to try to put more in evidence a direct relationship between clinical cases of antibiotic resistance in humans and animals.
This is a general problem of artificial intelligence: the machine produces outputs which cannot be mastered in a pertinent way by humans. So, now that you have a potential interesting and promising technology, the future is open for researchers restricting their vision to very precise and most pertinent questions in infectious diseases.
This is the reason why I indicate "not applicable" or "no answer" in the appreciation of your manuscript.
Response: Thank you for your comments. Our manuscript did not present the evidence of a direct relationship between clinical cases of antibiotic resistance in humans and animals. We will indicate the evidence on future manuscript.
Additionally, we double checked Figures, it looks quite confusing. I recommend to add a supplementary table for readers (Table S1), which is better to show the detailed value for each group.
…the representative ESBL-producing bacteria, Escherichia, Enterobacter, and Klebsiella, had a higher proportion ratio than at the other sampling sites (Fig. 2 and Table S1).
(Page 4, line 123).
Table S1: Summary of OTUs detected in CHROMagar ESBL culture samples from Indonesian river water.
(Page 8, line 224 to 225).
Round 2
Reviewer 1 Report
Comments and Suggestions for Authors
N/A
Comments on the Quality of English Language
N/A